# Relationship between Transition Shock, Professional Stressors, and Intent to Leave the Nursing Profession during the COVID-19 Pandemic

**DOI:** 10.3390/medicina59030468

**Published:** 2023-02-27

**Authors:** Dario Nakić, Ivana Gusar, Ivana Franov, Marijana Matek Sarić, Marija Ljubičić

**Affiliations:** 1General Hospital Zadar, Bože Peričića 5, 23000 Zadar, Croatia; 2Department of Health Studies, University of Zadar, Splitska 1, 23000 Zadar, Croatia; 3Department of Surgery, University Hospital Centre Split, Spinčićeva 1, 21000 Split, Croatia

**Keywords:** transition shock, nurse, COVID-19, stress, Croatia

## Abstract

*Background and Objectives:* Moving nurses to a COVID-19 department may cause the phenomenon of transition shock, which occurs when already employed nurses change jobs. A set of confusing and uncertain feelings arise due to the unfamiliar work environment, which may lead to their intention to leave the nursing profession. The aim of this study was to examine transition shock and the contribution of both the respondents’ characteristics and the presence of stressors to the occurrence of transition shock in nurses assigned to work in COVID-19 departments. *Materials and Methods*: A cross-sectional study with 120 nurses employed in COVID-19 departments was conducted. Several linear regression models were used to assess the association between transition shock, personal and professional COVID-19 stressors, and the intention to leave the nursing profession. *Results*: Nurses who intended to leave the profession showed higher transition shock and higher personal and professional stressors (*p* < 0.001). Female nurses had lower transition shock (β = −0.16; *p* = 0.036) and higher personal COVID-19 stressors (β = 0.27; *p* < 0.001). University education contributes to the lowering of nurses’ transition shock (β = −0.16; *p* = 0.038). Nurses who did not intend to leave the nursing profession had lower personal COVID-19 stressors (β = −0.15; *p* = 0.044). Transition shock was associated with personal COVID-19 stressors (β = 0.39; *p* < 0.001) and professional COVID-19 stressors (β = 0.29; *p* < 0.001), and vice versa. *Conclusions*: The phenomenon of transition shock was present after nurses transitioned to working in COVID-19 departments. Transition shock may cause more nurses to leave the profession, which may have a strong impact on the health system in many countries that are already facing a shortage of nurses. Additional education on and preparation for adapting to new working conditions with psychological support could have an influence by lowering the level of nurses’ transition shock.

## 1. Introduction

The COVID-19 pandemic caused by the SARS-CoV-2 virus is one of the greatest global threats that has occurred in the history of mankind [1,2]. Unlike the general population, healthcare workers employed in the hospital system are continuously exposed to a high risk of infection [3]. Due to the speed with which the virus spread, the functioning of health systems around the world was significantly changed, which additionally caused unprecedented pressure on healthcare systems and healthcare workers [3,4,5]. Healthcare workers had to quickly adopt new knowledge in order to provide effective health care to sick patients with the mandatory use of protective equipment such as masks, goggles, and protective suits [4,6]. The combination of a lack of healthcare workers, the workload, ethical decisions about the distribution of resources, and the risk of disease transmission impairs the physical and mental health of all healthcare workers [7,8]. Previous relevant studies highlight the presence of symptoms of impaired mental health in health workers in general, including nurses [8]. The already demanding circumstances were made even more difficult due to the redistribution of work and a change in the workplace, i.e., transitioning to work within a COVID-19 department [4], which also impairs the mental health of employees [9].

Moving nurses to a COVID-19 department may cause the phenomenon of transition shock, which occurs when already employed nurses change jobs, resulting in a set of confusing and uncertain feelings that arise due to the unfamiliar work environment corresponding to the shock of reality [10]. Transition is the process of moving from one department to another, and it is often associated with sizable changes in the nurses’ goals, roles, and responsibilities [11], which may cause anxiety, fear, panic, stress, and other different emotional conditions. Emotional regulation is characterized as the nurses’ ability to modify their emotional state in order to support adaptive purpose behaviors [12]. For example, it was shown that nurses who felt emotional weariness had trouble managing or controlling their emotions at work [12]. On the other hand, studies confirm that more adaptive coping and greater resilience lead to greater work satisfaction and reduced fatigue [13]. Authors Ko and Kim state that in the clinical environment emotional shock occurs in new nurses [14], while other authors believe that an emotional upheaval occurs in both newly employed nurses and those with work experience [15]. Although it is known that the competencies of experienced nurses facilitate aspects of the transition in overcoming the problems of working [15,16], in pandemic conditions, working with COVID-19 patients was a completely new professional experience for everyone.

Transition shock, also called reality shock, is made up of emotions that a nurse experiences when they starts working in a new environment, and it consists of emotional, physical, and sociocultural “feedback” that an individual shows when they experience a negative or unexpected event in an unfamiliar environment [10]. The reality shock theory focuses on the need for the socialization of nurses in a new work environment, where four phases are described: the honeymoon phase, the rejection/regression phase, the recovery phase, and the resolution phase [17]. The honeymoon phase is usually short-lived, or even skipped [10] in the case of nurses who already have work experience as a nurse. In the rejection/regression or shock phase, the newly employed nurse begins to doubt themselves and their abilities, and has a feeling of “drowning” or being “terrified” and “overtired” [10,17]. In this stage, job satisfaction is at its lowest level, which leads to demands for another job or, ultimately, leaving the nursing profession [17]. If, in this phase, the nurses have support and the available mechanisms for effectively adapting to the new role, they will successfully move to the recovery phase [18], which includes a sense of belonging to the new environment and the acceptance of one’s role as it develops [10]. The nurse feels a lower level of anxiety and has improved mechanisms for dealing with new situations [17]. The final phase is called the resolution phase, and is characterized by a decision about future: to stay in the nursing profession, change workplace and/or employer, or leave the nursing profession completely [10]. Some of the nurses effectively transition between roles and become confident and competent professionals, whereas the rest may experience burnout at work due to an ineffective transition and consequently leave the profession [17]. It is assumed that the reality shock theory represents a cyclical process and that individual nurses, after completing the resolution phase, can return to the shock phase when facing new challenges at work [10]. In these circumstances, the cycle can start from the beginning. It is likely that these complex circumstances occurred during the transition to work in COVID-19 departments.

Transitioning to work in COVID-19 departments may cause the nurses to worry over spreading the virus to their own family and, combined with other significant professional stressors at work, might make transitioning difficult, disturb professional harmony, and increase their desire to leave the nursing field. Studies confirm that during the COVID-19 pandemic, there was increased intent for the nurses to leave their profession [19,20,21]. However, previous studies related to the transition mostly identify moving from the role of a student to the role of an employee, where there is a mismatch between personal expectations and the reality in clinical practice [10,18,22]. Studies have also analyzed the experience of transition shock of nursing students in clinical practice, transition for new nurses [23], and transition for nurses when moving from a different department [11,14]; however, studies about the transition of nurses with existing experience to a COVID-19 department has not yet been represented. To our knowledge, no study has investigated the association between the level of transition shock and the nurses’ association with COVID-19 stressors and intent to leave their profession regarding nurses with existing working experience moving to a COVID-19 department.

Given the lack of available research on the potential presence of transition shock in nurses who replaced their previous workplace with work in COVID-19 departments, the aim of this study was to examine the occurrence of transition shock and the contribution of both the characteristics of the respondents and the presence of stressors to the occurrence of transition shock in nurses assigned to work in COVID-19 departments. We hypothesized that nurses, when replacing their previous department and moving to aCOVID-19 department, experience both transition shock and a personal and professional stressor associated with sociodemographic characteristics.

## 2. Participants and Methods

### 2.1. Participants

This cross-sectional study involved 120 employed nurses. The inclusion criteria included nurses who moved from their previous workplace and started working in a department caring for patients with COVID-19 during the pandemic. Nurses who did not change their previous work place and did not provide nursing care for COVID-19 patients were excluded. The study took place from June to August 2022 at the Clinic of Infectious Disease at the University Hospital of Split.

Nurses received the link for the Google form questionnaire via their e-mail addresses, and the research was completely anonymous. The e-mail address and telephone number of the examiner were listed in the questionnaire, and nurses could contact the principal investigator in case of any difficulties. The nurses gave informed consent by pressing a button to indicate an agreement to participate. The study was approved by the Ethical Committee of the University Hospital of Split (No. 2181-147/01/06/M.S.-22-03). The study was conducted according to principles of the Declaration of Helsinki.

### 2.2. Questionnaire

The sociodemographic characteristics of the nurses, such as gender, age, level of education, marital status, parenting, length of service, duration of nursing care provided for COVID-19 patients, and intention to leave the nursing profession, were measured.

The Transition shock scale (short form) was used to assess the nurses’ perceptions of transition shock. The scale includes 15 items that capture reality-shock-related issues and concerns among COVID-19 nurses. Nurses were asked to assess each item on a 4-point Likert scale ranging from 1 (does not apply to me at all) to 4 (applies to me completely). The items were classified into five dimensions: the nursing role, expectations, relationships, performance, and private life. Higher score indicates a higher level of transition shock (range 15–60) [24].

The COVID-19 questionnaire (short form), which contained 8 items, was used to assess personal and professional stressors during the provision of nursing care for patients with COVID-19. Items were scored on 4-point Likert-type scale (1—does not apply to me at all, 2—applies to me a little, 3—moderately applies to me, 4—applies to me a lot). A higher score indicates a higher level of personal (range 3–15) and professional (range 5–20) stressors [25].

### 2.3. Statistical Methods

The statistical analysis was conducted using SPSS 25.0 (IBM, Armonk, NY, USA). We used the Kolmogorov–Smirnov normality test to assess the data distribution, while Cronbach’s alpha was used for internal consistency. For descriptive statistics, we used the median and interquartile range, while absolute numbers and percentages were used for categorical variables. To analyze the difference in level of transition shock according to sociodemographic characteristics, the Mann–Whitney U test and the Kruskal–Wallis test were used. Several linear regression models were conducted to access associations between the dependent variables of transition shock, personal and professional stressors, and predictors. Independent variables were nurses’ sociodemographic characteristics, duration of providing nursing care for COVID-19 patients, and the intention to leave the nursing profession. Statistically significant values were indicated by *p* < 0.05.

## 3. Results

### 3.1. Sociodemographic Characteristics of Study Group

Out of 120 nurses, 76 (63.3%) were women and 44 (36.7%) were men. The average age of nurses was 27 years. Married nurses (57.5%) and nurses without children (64.2%) were dominant. More than half of the nurses were nurses for general care (53.3%), followed by nurses with Bachelor’s degrees (36.7%) and nurses with Master’s degrees (10%). A total of 77.5% of nurses had less than 10 years of work experience, and 58.2% had worked in the COVID-19 department for up to one year. The average level of transition shock was 38.0 (IQR 16.0), whereas the average levels for personal stressors and professional stressors were 14.0 (IQR 4.0) and 9.0 (IQR 4.0), respectively (Table 1).

### 3.2. Differences in Transition Shock and Perception of Personal and Professional Stressors

Differences in transition shock and professional COVID-19 stressors by gender, marital status, education, and length of service were not found. The female gender (Mdn = 14.0, IQR = 4.0; mean rank = 65.6; *p* = 0.034) and nurses who were older than 41 years (Mdn = 1.0, IQR = 4.0; mean rank = 62.6; *p* = 0.042) had the highest perceptions of personal stressors. Nurses who have an intention to leave the nursing profession had higher transition shock (Mdn = 43.0, IQR = 11.0; mean rank = 74.4; *p* < 0.001), higher personal COVID-19 stressors (Mdn = 15.0, IQR = 5.0; mean rank = 76.6; *p* < 0.001), and higher professional COVID-19 stressors (Mdn = 10.0, IQR = 3.0; mean rank = 77.8; *p* < 0.001) (Table 2).

### 3.3. Associations between Transition Shock and Stressors during Nurses’ Work in a COVID-19 Department

Linear regression models confirm the association between the female gender and lower transition shock (β = −0.16; *p* = 0.034), whereas the female gender contributes to a higher perception of personal COVID-19 stressors (β = 0.27; *p* < 0.001). A higher level of education contributes to lower transition shock (β = −0.16; *p* = 0.038). Nurses with no intentions to leave the nursing profession had a significantly lower perception of personal COVID-19 stressors (β = −0.15; *p* = 0.044). Transition shock statistically and significantly contributes to personal (β = 0.39; *p* < 0.001) and professional (β = 0.29; *p* = 0.002) stressors. Personal stressors are positively associated with transition shock (β = 0.44; *p* < 0.001) and professional stressors (β = 0.36; *p* < 0.001), and vice versa (Table 3).

## 4. Discussion

The aim of this study was to examine the transition shock and the contribution of both the respondents’ characteristics and the presence of stressors to the occurrence of transition shock in nurses assigned to work in COVID-19 departments. As we hypothesized, when nurses replaced their previous department with a COVID-19 department, they experienced a transition shock and felt personal and professional stressors. Although the studies mainly state that the transition shock occurs when the nurses move from the role of a student to the role of an employee [10], our results indicate that the nurses also experience transition shock when they change workplace. This was particularly emphasized during the COVID-19 pandemic. Many healthcare workers were faced with a change in their primary workplace, as well as a change in the execution of work procedures and the adoption of new protocols and procedures when performing work tasks. It was necessary for some employees to be assigned to newly formed departments and workplaces that involved the direct care and treatment of COVID-19-positive patients [4].

In this study, there was a prevalence of younger nurses with less working experience, which may have contributed to transition shock [15]. Earlier research also confirmed that younger nurses are more susceptible to developing signs of transition shock [3,15]. Similar observations were reported in South Korea, where most of the investigated nurses complained about a lack of specific medical knowledge and skills required for the care of these patients [26]. Furthermore, their work was hindered by increased physical, psychological, and social pressure, largely caused by the fact that they were fighting against an insufficiently known virus, leading to uncertainty and fear, which are both extremely strong stressors [27]. Furthermore, studies conducted in Bangladesh also highlight greater difficulties in health workers that are female and belong to a younger age group [3,8]. These circumstances may causes an increased sensitivity to presenting personal and professional challenges and problems [28]. The already demanding circumstances are made even more difficult due to the redistribution of work and a change in workplace, i.e., moving to work within a COVID-19 department, which may impair the mental health of employees [4,9]. If we take into account that health, in addition to belonging to an individual anatomical–functional entity, also belongs to the narrative of “how I am/how I feel” [29], it is quite clear that newly created circumstances can greatly affect the health of employees in a pandemic.

The highest level of transition shock was recorded among nurses who intend to leave the nursing profession, which is one of the most concerning results of this study. It was noticed that, during the COVID-19 pandemic, a nurses’ wish to leave their profession rose approximately 5% more than in the period before the COVID-19 pandemic [30]. There may have been a heightened overload during the pandemic brought on by personal anxiety for family members and work-related pressures [21]. A higher transition shock in nurses who intend to leave their profession may be explained by a professional overload, inadequate working aspects, and higher personal and professional COVID-19 stressors, which all have an impact on their personal life. This could be a result of the transition or a toxic workplace environment, endangering the nurse’s well-being and disrupting their mental health [20]. However, inadequate working conditions and aspects of the organizational climate are cited by more than half of nurses who say they wished to leave nursing prior to the pandemic [31].

Namely, many studies have confirmed that healthcare workers are increasingly exposed to stress and a high workload and are worried about their families and their own future [31]. Furthermore, Zhang et al. pointed out that there are several different predictors of the health status of healthcare workers during the pandemic and warned of the need for individualization and support due to the diversity of healthcare systems facing the pandemic [25]. Therefore, it is very important to provide psychological support for nurses when working with COVID-19 patients, which can reduce transition shock, improve emotional regulation, and keep nurses in the profession. It is important for the nurses to improve their emotional regulation in order to manage their emotions during the internal process of rationalization and to be able to be aware of their own strengths and the possibility of resolving problems. An increased work satisfaction, improved cohesion, strengthening of resilience, cognitive coping, and mental health are some of the outcomes of emotion regulation for nurses [32]. These emotional regulation strategies, combined with other strategies in daily life, such as resting in nature, keeping a diary, meditation, group support, physical exercise, and other healthy lifestyle habits, certainly contribute to a reduction in the burnout of nurses [13].

Further, contrary to the results of the studies from Bangladesh [3,8], in our study, older female respondents experienced a higher level of personal COVID-19 stressors, which is probably the result of caring for their family. Similar results were reported in a qualitive study from Iran [33]. Nurses who worked with COVID-19 patients felt fear regarding transmitting the virus to family members, and were usually distant from them [33], which could increase the level of personal stress. Similar results were recorded with German healthcare workers, who also expressed an increased level of stress and concern for their future and their families, as well as an increase in their subjective burden [31].

As expected, nurses who have an intention to leave the nursing profession reported a higher level of transition shock and personal and professional COVID-19 stressors. The balance between private life and work is seriously disturbed by working in the conditions of the pandemic [34]. A satisfactory balance between private life and work during the pandemic is related to the psychological state of healthcare workers [35]. Therefore, it can be assumed that these aforementioned factors also affected the experienced level of transition shock among our respondents. They were worried about their health and the health of their family, and were not able to meet the needs of their children and other family members [36]. Nevertheless, they faced new and unfamiliar working circumstances [37], and many nurses around the world were infected, hospitalized, or even died [38]. Previous studies also show that health workers involved in providing health care during epidemics are at a high risk of developing mental health problems [39,40]. In the reports from Wuhan, the center of the COVID-19 pandemic, both physiological support and previous experience in working with patients with infectious diseases correlated with low levels of concern in confronting COVID-19 [41].

In our study, the level of transition shock was inversely proportional to the level of the respondents’ education. These results are similar to the results of earlier research that also stated that the level of competence is important in adapting to the new work environment and experiencing transition shock [14,37].

The results from the linear regression analysis additionally indicate a connection between the level of transition shock and personal and professional COVID-19 stressors. A more pronounced experience of personal and professional COVID-19 stressors resulted in a higher level of transition shock and vice versa. The results of other studies also point to the connection between stress and transition shock [10,18,33]. As Lina and Setiawan state, transition or culture shock is a set of confused or uncertain emotions experienced by an individual in an unfamiliar environment [42]. Considering that COVID-19 was a completely unknown disease [37], it is logical that a certain level of personal and professional stress and transition shock occurred in nurses when moving to a new workplace due to the care of sick patients.

Despite its advantages, this study had some limitations. First, this was a cross-sectional study, and proving the actual causality of the transition shock in nurses that were moved to the COVID-19 department is not possible. Second, the study only included one hospital, without a control group, making it difficult to compare results. Additionally, a relatively small sample size may affect the statistical power and detection of the association between transition shock and predictors. Unfortunately, at the time of the study, we were not able to recruit more nurses and include a control group. Furthermore, although previous studies emphasize the importance of economic status on the incidence of mental health problems [3,8], data on the economic status and potential economic limitations were not collected in this study.

Despite these limitations, the results of this research can help the administrations of health institutions around the world to better understand the severity of the circumstances in which nurses find themselves when transferring from their previous workplace to a COVID-19 workplace. Considering the results, it is possible to include different types of interventions that would reduce nurses’ sensitivity to personal and professional stress, as well as the level of transition shock. This includes supporting nurses during the transition, the implementation of staff training and mentor support, strengthening nurses’ professional relationships, promoting nurses’ autonomy, awards for quality work performed, the possibility of using more extended holidays, and increasing the system’s efficiency and work conditions, which will all contribute to facilitating nurses in coping with the transition.

## 5. Conclusions

Transition shock, as a special phenomenon during any part of the professional career of a nurse, should not be neglected. In this study, the appearance of transition shock was also present during COVID-19 infection due to its status as an unknown threat, the separation of nurses from their families, and their worry for their own health and the health of their families and friends. Transition shock has a great influence on the nurses’ quality of work, satisfaction with their own professional performance, and their confidence, but also has an impact on their intention to leave the nursing profession. Additionally, these circumstances may have negative impacts on the nurses’ health. This may have a great negative consequence on the health system, which is already faced with a lack of nurses. Furthermore, scientists warn that COVID-19 will unfortunately not be the last pandemic in our lifetime, and the threat posed by zoonoses, which are infectious diseases that are transmitted from animals to humans, is on the rise. Therefore, this research, research similar to this, and their results are very important for all communities as they could provide a platform for creating policies and/or ultimately provide the tools to effectively support this requirement in the future. Improving professional practice and additional education on and preparation for adapting to new working conditions with psychological support could have an influence on a more successful adaptation to and lower level of transition shock, professional stressors, and nurses’ intention to leave their profession.

## Figures and Tables

**Table 1 medicina-59-00468-t001:** Sociodemographic characteristics of study group; N = 120.

Age (years), Mdn (IQR)	27.0 (7.0)
Age groups; N (%)	
20–30 years	86 (71.7%)
31–40 years	27 (22.5%)
≥41 years	7 (5.8%)
Gender; N (%)	
Female	76 (63.3%)
Male	44 (36.7%)
Marital status, N (%)	
Married	69 (57.5%)
Single	48 (40.0%)
Divorced	3 (2.5%)
Have children, N (%)	
Yes	43 (35.8%)
No	77 (64.2%)
Education level, N (%)	
High school	64 (53.3%)
Bachelor’s	44 (36.7%)
Master’s	12 (10.0%)
Length of service, N (%)	
<10 years	93 (77.5%)
>10 years	27 (22.5%)
Working in a COVID-19 department, N (%)	
<1 year	70 (58.3%)
>1 year	50 (41.7%)
Leaving the nursing profession, N (%)	
Intention	47 (39.2)
No intention	73 (60.8)
Transition shock (sum)	38.0 (16.0)
Personal COVID-19 stressors (sum)	14.0 (4.0)
Professional COVID-19 stressors (sum)	9.0 (4.0)

Note: Mdn = Median; IQR = Interquartile Range; Transition shock (0–120); Personal COVID-19 stressors (0–120); Professional COVID-19 stressors (0–120).

**Table 2 medicina-59-00468-t002:** Differences in transition shock and personal and professional COVID-19 stressors by sociodemographic chrematistics in a sample of nurse (N = 120).

	Transition Shock			Personal COVID-19 Stressors			Professional COVID-19 Stressors		
	Mdn (IQR)	Mean Rank	*p*	Mdn (IQR)	Mean Rank	*p*	Mdn (IQR)	Mean Rank	*p*
Age									
20–30 years	38.0 (17.0)	59.3	0.403 *	14.0 (4.0)	62.6	0.042 *	9.0 (4.0)	56.7	0.159 *
31–40 years	38.0 (16.0)	60.0		12.0 (4.0)	48.3		9.0 (5.0)	69.7	
≥41 years	43.0 (12.0)	77.6		15.0 (4.0)	82.0		9.0 (2.0)	71.6	
Gender									
Female	39.0 (16.0)	59.5	0.693 *	14.0 (4.0)	65.6	0.034 *	9.0 (5.0)	59.9	0.811 *
Male	38.0 (13.0)	62.2		12.0 (5.0)	51.7		9.0 (3.0)	61.5	
Marital status									
Married	38.0 (18.0)	61.6	0.638 *	14.0 (5.0)	64.1	0.186 *	9.0 (4.0)	65.4	0.068 *
Single, divorced	38.0 (15.0)	59.1		13.0 (4.0)	55.6		9.0 (5.0)	53.8	
Have children									
Yes	38.0 (17.0)	58.00	0.556 *	13.0 (4.0)	56.78	0.379 *	9.0 (4.0)	64.91	0.295 *
No	39.0 (16.0)	61.90		14.0 (4.0)	62.58		9.0 (4.0)	58.04	
Education									
High school	40.0 (15.7)	65.1	0.084 †	14.0 (4.0)	61.6	0.922 †	9.0 (5.0)	60.0	0.576 †
Bachelor’s	36.6 (13.8)	51.4		13.5 (4.8)	58.8		9.0 (3.8)	58.6	
Master’s	40.5 (9.5)	69.3		13.5 (4.0)	60.9		9.0 (3.3)	70.2	
Length of service									
<10 years	38.0 (16.0)	59.9	0.732 *	14.0 (3.5)	61.7	0.487 *	9.0 (4.0)	58.6	0.267 *
>10 years	39.0 (19.0)	62.5		13.0 (4.0)	56.4		9.0 (4.0)	67.0	
Working in a COVID-19 department									
<1 year	39.0 (15.0)	63.7	0.231 *	14.0 (3.0)	61.8	0.632 *	9.0 (4.0)	62.7	0.406 *
>1 year	37.0 (16.5)	56.0		13.0 (4.3)	58.7		9.0 (5.0)	57.4	
Leaving the nursing profession									
Intention	43.0 (11.0)	74.4	<0.001 *	15.0 (5.0)	76.0	<0.001 *	10.0 (3.0)	77.8	<0.001 *
No intention	36.0 (13.5)	51.6		12.0 (5.0)	50.5		8.0 (3.0)	49.3	

Note: * Mann–Whitney U test; † Kruskal–Wallis test.

**Table 3 medicina-59-00468-t003:** Associations between nurses’ sociodemographic characteristics, personal and professional COVID-19 stressors, and transition shock during work in a COVID-19 department using linear regression models (N = 120).

	Transition Shock			Personal COVID-19 Stressors			Professional COVID-19 Stressors		
	β	t	*p*	β	t	*p*	β	t	*p*
Age	0.06	0.49	0.624	−0.01	−0.11	0.913	0.15	1.23	0.222
Gender (male was reference group)									
Female	−0.16	−2.14	0.034	0.27	4.01	<0.001	−0.08	−1.06	0.291
Marital status (single/divorces was reference group)									
Married	−0.03	−0.31	0.757	0.14	1.77	0.079	0.03	0.35	0.726
Children (yes was reference group)									
No	0.09	0.90	0.368	0.09	0.96	0.341	−0.02	−0.21	0.838
Education (high school was reference group)									
University	−0.16	−2.10	0.038	0.01	0.12	0.909	0.04	0.50	0.617
Length of services (<10 years was reference group)									
>10 years	0.07	0.61	0.543	−0.12	−1.21	0.228	0.01	0.10	0.918
Working in a COVID-19 department (<1 year was reference group)									
>1 year	−0.09	−1.18	0.241	0.02	0.31	0.759	−0.04	−0.48	0.631
Leaving the nursing profession (intention was reference group)									
No intention	−0.05	−0.65	0.518	−0.15	−2.04	0.044	−0.13	−1.63	0.106
Transition shock (sum)	-	-	-	0.39	4.71	<0.001	0.29	3.13	0.002
Personal COVID-19 stressors (sum)	0.44	4.71	<0.001	-	-	-	0.36	3.71	<0.001
Professional COVID-19 stressors (sum)	0.28	3.13	0.002	0.31	3.71	<0.001	-	-	-

Note: β—beta coefficient; *p*—*p*-value.

## Data Availability

All data are available from the corresponding author upon reasonable request.

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
