# Peer review of "Relationship between Transition Shock, Professional Stressors, and Intent to Leave the Nursing Profession during the COVID-19 Pandemic"

_medicina, 2023, doi:10.3390/medicina59030468_

Round 1

Reviewer 1 Report

This is a good study. However, I have some minor change requests. Please provide more references, if possible, both in the introduction and discussion. These study findings are in line with the study you have done: https://doi.org/10.1016/j.heliyon.2021.e06715; https://doi.org/10.1016/j.heliyon.2021.e06985; https://doi.org/10.1016/j.dialog.2022.100037. If possible, you can read and cite this similar studies. All the studies related to COVID-19 and the mental health of healthcare professionals, including nurses. 

Best of luck with your revised version of this good piece of work. 

Author Response

Dear reviewer,

thank you for your prompt response to the revision of the manuscript: Relationship between transition shock, professional stressors, and intent to leave the nursing profession during a pandemic Covid-19. (medicina-2208709). All changes in the manuscript are written in red. The paper has been improved by taking into account the comments we received from the reviewers. All the concerns have been carefully addressed. A proofreading was performed throughout the manuscript by a MDPI English editing servis.

Reviewer 2 Report

The research is very interesting anche could be improve in other hospital, as you wrote. I think that is important to explore (in the introduction and in the discussion) the link between the transition shock anche emotional regulation, as some results (by literature and your research) highlight.

Then, in the discussion, you could deepen the transition shock impact on health in general (eg. Turchi, G. P., et al. (2022). A contribution towards health. Journal of Evaluation in Clinical Practice28(5), 717) and in other areas (work transition, environment transition etc).

Author Response

(The authors gave the same response as above.)

Round 2

Reviewer 1 Report

I do not have any more comments.